# Clinical value of different anti-D immunoglobulin strategies for preventing Rh hemolytic disease of the fetus and newborn: A network meta-analysis

**Xiaohui Xie**[1☯], **Qiurong Fu**[2☯], **Ziwei Bao**[3☯], **Yi Zhang**[4]*, **Dan Zhou**[1]*

1 Department of Obstetrics and Gynecology, the First People's Hospital of Neijiang, Neijiang, Sichuan Province, P. R. China, 2 Department of Nursing, The first Affiliated Hospital of Hainan Medical University, Haikou, Hainan Province, P. R. China, 3 Department of medicine, Southwest Medical University, Luzhou, Sichuan Province, P. R. China, 4 Department of General Surgery, the First People's Hospital of Neijiang, Neijiang, Sichuan Province, P. R. China

☯ These authors contributed equally to this work.
* 307910985@qq.com (YZ); 82150436@qq.com (DZ)

**Data Availability Statement:** All relevant data are within the manuscript and its Supporting Information files.

## Abstract

### Background

Several anti-D immunoglobulin strategies exist for preventing Rh hemolytic disease of the fetus and newborn. This study systematically assessed the clinical value of those therapeutic strategies.

### Methods

The Web of Science, PubMed, EMBASE, China National Knowledge Infrastructure (CNKI) and Wanfang databases were searched for eligible studies that evaluated the value of different anti-D immunoglobulin strategies in preventing maternal anti-D antibody sensitization. Combined odds ratios (ORs) and their 95% confidence intervals (CIs) were calculated. The network meta-analysis was conducted using Stata 14.2 and WinBUGS 1.4.3 software.

### Results

Twenty-four original studies involving 64860 patients were included. Among all therapeutic measures, injecting 300 μg anti-D immunoglobulin at 28 and 34 gestational weeks (antenatal 5/E) appeared to be the most effective measure for preventing maternal antibody sensitization (surface under the cumulative ranking curve [SUCRA] = 96.8%), while a single injection at 28 gestational weeks (SUCRA = 89.2%) was the second most effective. Administering no injection or a placebo (SUCRA = 0.0%) was the least effective intervention measure.

### Conclusion

Among the therapeutic measures, antenatal 5/E appeared to be the best method for reducing the positive incidence of anti-D antibodies in the maternal serum; thus, it may be the most effective treatment for preventing fetal hemolytic disease.

**Funding:** The author(s) received no specific funding for this work.

**Competing interests:** The authors have declared that no competing interests exist.

## Introduction

Hemolytic disease of the fetus and newborn (HDFN) can lead to fetal hemolytic anemia, jaundice, intellectual retardation, premature birth, abortion and stillbirth. HDFN is an important cause of neonatal morbidity and death [1–3]. To reduce the incidence of HDFN and mortality among fetuses and neonates, anti-D immunoglobulin has been tested in clinical trials in the United Kingdom and United States since the 1960s. Anti-D immunoglobulin has been used to prevent postpartum disease in RhD-negative women and has greatly reduced HDFN-related morbidity as well as fetal and neonatal mortality [4]. The anti-D antibody production rate in the maternal serum after immunization has also decreased significantly from 12–13% to approximately 1.2%. Prenatal prophylaxis with anti-D immunoglobulin in Rh-negative mothers can further reduce anti-D antibody production in maternal sera, which has further reduced the incidence of hemolytic diseases in fetuses and newborns since 1980 [5–11].

However, multiple countries recommend various anti-D immunoglobulin injection schemes, and no consensus has been reached on the use of anti-D immunoglobulin worldwide. Routine prenatal anti-D prophylaxis (RAADP) is recommended in some countries, while postpartum anti-D immunoglobulin injections are still used in other countries. Furthermore, the injection dose differs in some countries due to the lack of available immunoglobulin. Lee et al. suggested that administering low doses of anti-D immunoglobulin (50 μg) provided no benefit [12]. However, excessive doses may increase the risk of allergic reactions and infectious diseases.

Until now, no meta-analysis has been conducted to evaluate the association between anti-D antibody production rates in the maternal serum and various therapeutic strategies regarding anti-D immunoglobulin. We conducted this study to systematically evaluate the preventive effects of anti-D immunoglobulin on HDFN via network meta-analysis based on all related published data.

## Material and methods

### Search strategy

A comprehensive search strategy was employed to search the PubMed, EMBASE, Web of Science, China National Knowledge Infrastructure (CNKI) and Wanfang databases. The latest search was conducted on 7 July 2019. The following keywords were used in accordance with the search strategy: "RhD-negative" OR "D-negative" OR "Rh(D) Immuno-Globulin" OR "Anti-D Immunoglobulin" OR "Anti-D Antibody" OR "the hemolytic disease of the newborn" OR "haemolytic disease of the newborn" OR "HDFN" et al.

### Inclusion and exclusion criteria

The inclusion criteria were as follows: 1) randomized controlled studies on administering anti-D immunoglobulin injections to RhD-negative pregnant women; 2) Rh-positive fetuses in intrauterine pregnancies of Rh-negative pregnant women; 3) reported dose and frequency of anti-D immunoglobulin injections; and 4) reported positive incidence of anti-D antibody in postpartum mothers. Duplications, reviews, case reports, conference abstracts, and studies without useful data were excluded.

### Study selection

Two authors (XXH and FQR) screened the abstracts and titles of eligible publications and judged whether to further review the full text independently. We contacted the trial author when full texts were unavailable. Full texts were independently reviewed by XXH and ZD. In

the case of any disagreement during the selection process, the decisions were obtained after group discussion. Finally, we used flow chart to show the total number of retrieved references and the number of included and excluded studies.

### Data extraction

Two investigators collected data independently in accordance with predesigned tables, which included the name of the first author, publication year, country, sample size, intervention measures, control measures, and anti-D antibody production rate in the maternal serum.

Two researchers independently assessed the quality of all included studies using the Newcastle-Ottawa quality assessment scale (NOS). This method comprised three parameters of quality: selection (score: 0–4), comparability (score: 0–2), and outcome assessment (score: 0–3), with total scores ranging from 0–9. NOS scores >6 were considered to indicate high-quality studies.

### Statistical analysis

Stata statistical software, version 14.2 and WinBUGS 1.4.3 were applied to analyze the relationship between anti-D antibody production rates in the maternal serum and various anti-D immunoglobulin injection regimens. The random-effects model with vague priors for multi-arm trials was used. The model parameters were estimated using the Markov chain Monte Carlo method of Gibbs sampling. The results are reported as the odds ratio (OR) and standardized mean difference(SMD) with 95% confidence intervals (CIs). To evaluate the inconsistency between direct and indirect effect estimates for the same comparisons, we evaluated each closed loop in the network. In a closed loop, we employed the inconsistency factor (IF) to evaluate heterogeneity among the included studies. Node analysis showed that the direct and indirect comparisons of each node did not differ (P>0.05), and the consistency model was used for convergence. To rank the treatments, we used the surface under the cumulative ranking probabilities (SUCRA). A comparison-adjusted funnel plot was used to assess the presence of small-study effect and publication bias.

## Results

### Characteristics of eligible studies

Fig 1 shows the literature retrieval procedure. After further discussing and considering the retrieved articles, 24 eligible articles[5,8,10–31] were ultimately identified, and 64860 patients were included in this network meta-analysis, with an average sample size of 2702.5 (range 54–12825). Among those studies, nine intervention-measure dosages for administering anti-D immunoglobulin were analyzed: 250 μg within 28 gestational weeks (antenatal 1/A), 300 μg within 28 gestational weeks (antenatal 2/B), 50 μg within 28 and 34 gestational weeks (antenatal 3/C ), 100 μg between 28 and 34 gestational weeks (antenatal 4/D), 300 μg between 28 and 34 gestational weeks (antenatal 5/E), placebo or blank control group (blank/F),100 μg ≤ dosage < 200 μg within 72 h postpartum (postnatal 1/G), 200 μg ≤ dosage < 300 μg within 72 h postpartum (postnatal 2/H), and 300 μg ≤ dosage < 500 μg within 72 h postpartum (postnatal 3/I). All articles were written in good-quality English. **Table 1** summarizes the main characteristics of all included cohort studies. Table 2 has the treatment abbreviations.

### Network meta-analysis results

**Network relationship and inconsistency test.** In this network meta-analysis, the association between various anti-D immunoglobulin strategies and their clinical value in HDFN was

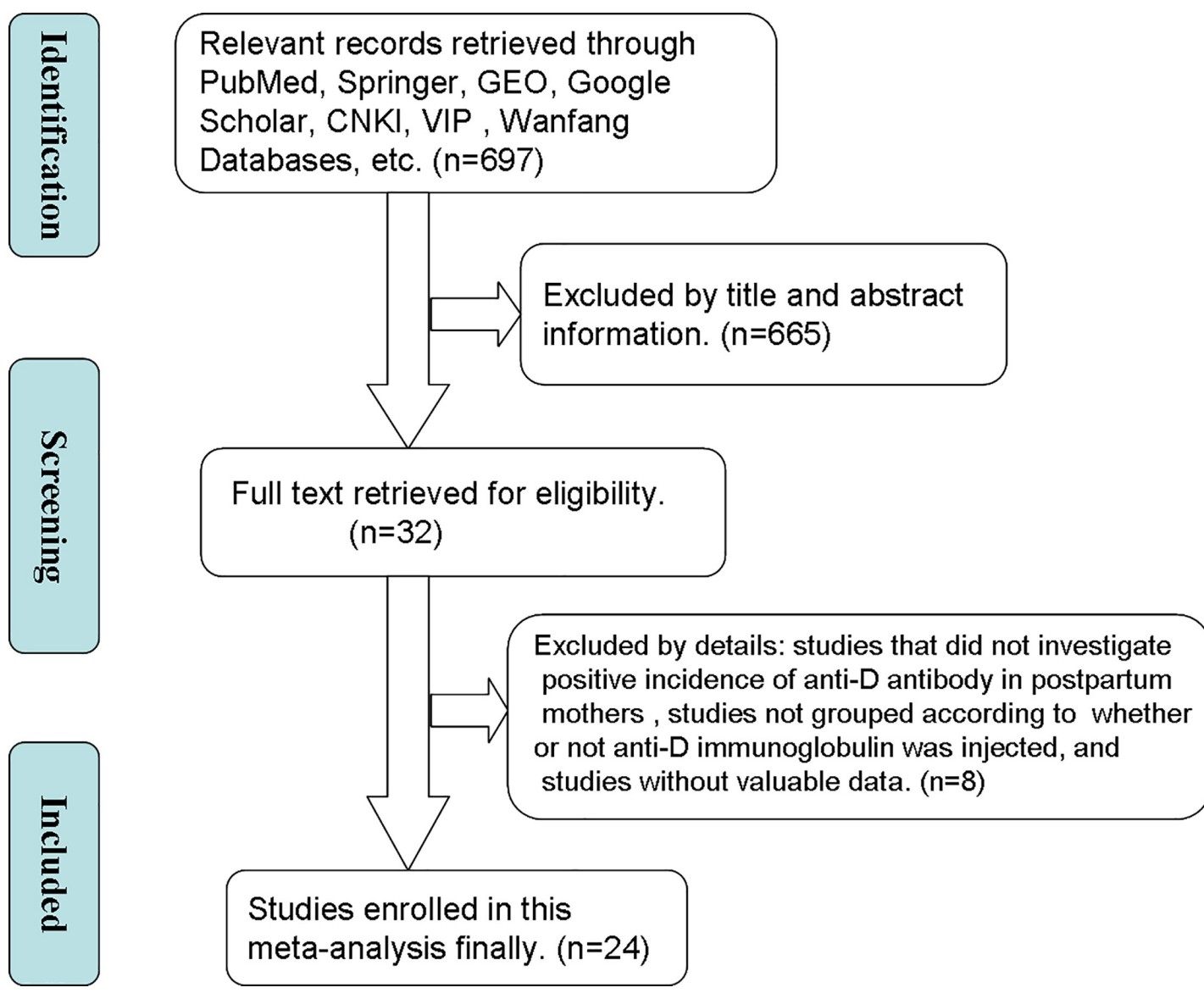

**Fig 1. Flow diagram of included studies.**

analyzed for 24 cohort studies comprising 64860 patients and nine treatment measures. Fig 2 shows the network relationship among the different treatment measures. Nodes are proportional to the number of patients included in the corresponding treatments, and edges are weighted according to the number of studies included in the respective comparisons. Nine treatment measures formed three triangles and two quadrilateral closed loops. The inconsistency factor was obtained under the inconsistency model using Gemetc software. Fig 3 shows the inconsistency plot used to identify heterogeneity among studies in the closed loop of this network meta-analysis. Three triangular loops and two quadratic loops are present in the network meta-analysis. The results showed that the inconsistency factor (IF) was $0.11 \leq 2.13$, and the 95% confidence interval (CI) contained 0, suggesting that statistical inconsistency may not exist among these five closed loops (Fig 3). Furthermore, node analysis was used to analyze the differences between direct and indirect comparisons among treatment measures (Table 3).

**Table 1. Main characteristics of all included studies.**

| First author | Year | Country | Sample size | I/C | Intervention | Control | Multivariate analysis | NOS |
|---|---|---|---|---|---|---|---|---|
| Ascari WQ | 1968 | America | 1280 | 781/499 | **postnatal 3/I** | **blank/F** | YES | 7 |
| Ascari WQ | 1969 | America | 2876 | 1834/1042 | **postnatal 3/I** | **blank/F** | YES | 8 |
| Bryant EC | 1969 | America | 355 | 191/164 | **postnatal 3/I** | **blank/F** | NO | 8 |
| Jennings ER | 1968 | Canada | 493 | 258/235 | **postnatal 3/I** | **blank/F** | NO | 7 |
| Pollack W | 1968 | America | 1286 | 787/499 | **postnatal 3/I** | **blank/F** | NO | 8 |
| Robertson JG | 1969 | Scotland | 212 | 100/112 | **postnatal 3/I** | **blank/F** | NO | 7 |
| Stenchever MA | 1971 | America | 54 | 26/28 | **postnatal 3/I** | **blank/F** | NO | 7 |
| White CA | 1970 | America | 5438 | 3784/1654 | **postnatal 3/I** | **blank/F** | NO | 8 |
| Dudok D | 1968 | Holland | 662 | 333/329 | **postnatal 3/I** | **blank/F** | NO | 9 |
| Clake CA | 1968 | England | 197 | 95/102 | **postnatal 3/I** | **blank/F** | NO | 9 |
| Buchanan DI | 1969 | Canada | 430 | 223/207 | **postnatal2/H** | **postantal 1/G** | NO | 9 |
| Chown B | 1969 | Canada | 2216 | 358/500;858/500 | **postnatal1/G; postantal3/I** | **blank/F** | NO | 8 |
| John GC | 1969 | Canada | 202 | 65/42;53/42 | **postnatal1/G; postantal3/I** | **blank/F** | NO | 9 |
| Tovey LA | 1983 | England | 9178 | 3875/5303 | **antenatal 4/D** | **postantal 1/G** | NO | 7 |
| Huchet J | 1987 | France | 1189 | 599/590 | **antenatal 4/D** | **postantal 1/G** | YES | 8 |
| Bowam JM | 1987 | Canada | 12836 | 9303/3533 | **antenatal 2/B** | **postantal 3/I** | NO | 6 |
| Trolle B | 1989 | Denmark | 700 | 354/346 | **antenatal 2/B** | **postantal 2/H** | NO | 8 |
| Mayne S | 1997 | England | 2851 | 1425/1426 | **antenatal 4/D** | **postantal 3/I** | NO | 9 |
| Mackenzie IZ | 1999 | England | 6466 | 3320/3146 | **antenatal 4/D** | **postantal 3/I** | NO | 9 |
| Mackenzie IZ | 2004 | England | **491** | **248/243** | **antenatal 2/B** | **postantal 3/I** | NO | 9 |
| Lee D | 1995 | England | **1180** | **648/532** | **antenatal 3/C** | **blank/F** | YES | 8 |
| Bowam JM | 1978 | Canada | 2361 | 2109/252 | **antenatal 5/E** | **antenatal 2/B** | NO | 7 |
| Bowam JM | 1978 | Canada | 2612 | 1598/1014 | **antenatal 2/B** | **postantal 3/I** | NO | 7 |
| Hermann M | 1984 | Sweden | 9295 | 4895/4400 | **antenatal 1/A** | **postantal 2/H** | NO | 7 |

*P*>0.05 indicates that no statistical inconsistencies were observed, suggesting that a network meta-analysis can be accomplished by directly or indirectly comparing different therapeutic measures. Thus, data on various treatment measures can be converged using consistency models.

**Results of the Bayesian network meta-analysis.** According to the network of comparisons (Table 4), the antenatal 5/E, antenatal 2/B, antenatal 4/D, antenatal 1/A, postnatal 3/I, postnatal 2/H, and antenatal 3/C regimens significantly reduced the serum anti-D antibody-positive rates compared with that of the blank/F regimen alone (antenatal 5/E vs. blank/F: OR = 0.00, 95% CI = 0.00–0.04; antenatal 2/B vs. blank/F: OR = 0.01, 95% CI = 0.00–0.01; antenatal 4/D

**Table 2. Treatment abbreviations.**

| Full name | Abbreviations |
|---|---|
| Administered 250 μg within 28 gestational weeks | antenatal 1/A |
| Administered 300 μg within 28 gestational weeks | antenatal 2/B |
| Administered 50 μg within 28 and 34 gestational weeks | antenatal 3/C |
| Administered 100 μg between 28 and 34 gestational weeks | antenatal 4/D |
| Administered 300 μg between 28 and 34 gestational weeks | antenatal 5/E |
| Placebo or blank control group | blank/F |
| Administered 100 μg ≤ dosage < 200 μg within 72 h postpartum | postnatal 1/G |
| Administered 200 μg ≤ dosage < 300 μg within 72 h postpartum | postnatal 2/H |
| Administered 300 μg ≤ dosage < 500 μg within 72 h postpartum | postnatal 3/I |

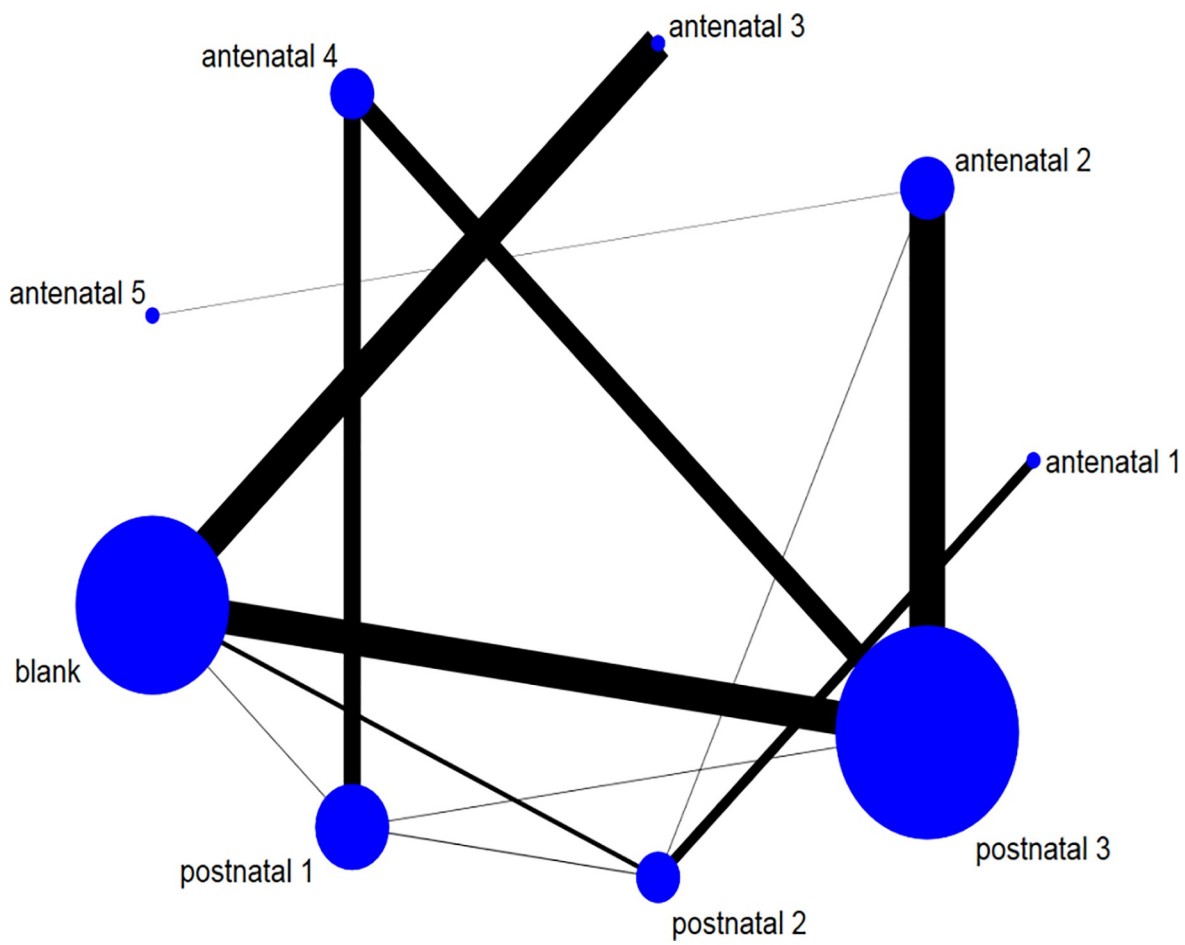

**Fig 2. Network relationship of the included treatment measures.**

vs. blank/F: OR = 0.01, 95% CI = 0.01–0.03; antenatal 1/A vs. blank/F: OR = 0.05, 95% CI = 0.01–0.18; postnatal 3/I vs. blank/F: OR = 0.04, 95% CI = 0.02–0.06, *P*<0.05; postnatal 2/ H vs. blank/F: OR = 0.11, 95% CI = 0.04–0.31; antenatal 3/C vs. blank/F: OR = 0.15, 95% CI = 0.09–0.24; all P<0.05). This indicated that injections of anti-D immunoglobulin, whether before or after delivery, significantly reduced the incidence of maternal serum that was positive for anti-D antibody in Rh-negative mothers with Rh-positive fetuses. Moreover, antenatal 5/E, antenatal 2/B, antenatal 4/D, antenatal 1/A and postnatal 3/I were the most effective treatment measures for reducing the incidence of maternal anti-D antibody positivity (antenatal 5/E vs. antenatal 2/B: OR = 0.12, 95% CI = 0.00–6.05; antenatal 2/B vs. antenatal 4/D: OR = 0.41, 95% CI = 0.20–0.82, *P*<0.05; antenatal 4/D vs. postnatal 3/I: OR = 0.39, 95% CI = 0.22–0.67, P<0.05; postnatal 3/I vs. antenatal 1/A: OR = 0.78, 95% CI = 0.20–3.08, *P*>0.05). Similarly, we used a forest plot to represent the network meta-analysis results (Fig 4). All immunization schemes were significantly more effective than was the blank control scheme (*P*<0.05).

**Rank probability.** Injecting 300 μg of anti-D immunoglobulin at 28 and 34 gestational weeks (antenatal 5/E) was the most effective treatment (surface under the cumulative ranking curve [SUCRA] = 96.8%; Fig 5), and administering 300 μg within 28 gestational weeks (antenatal 2/B) was the second most effective (SUCRA = 89.2%). Administering no injection or a placebo was the least effective (SUCRA = 0.0%).

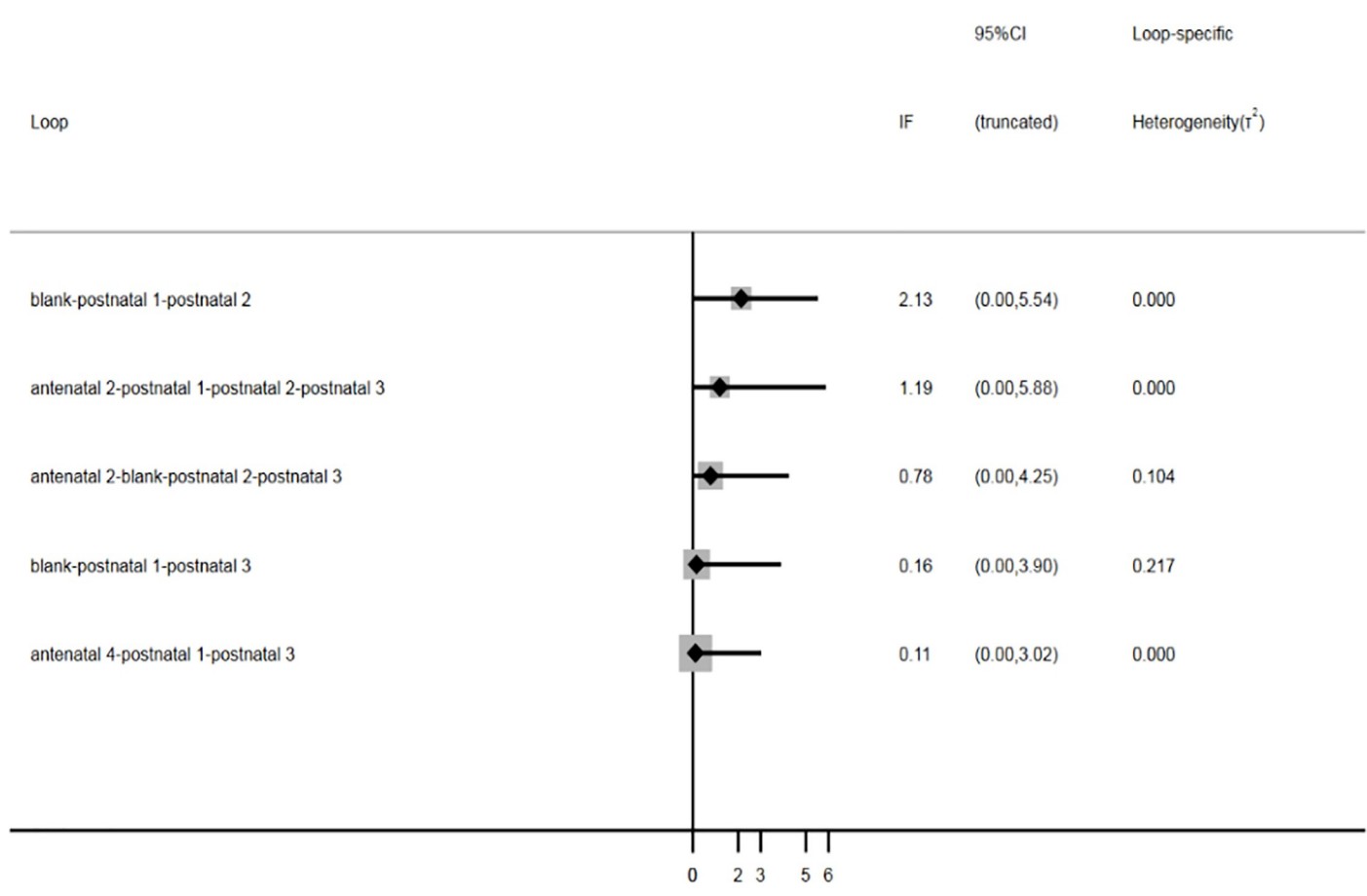

**Fig 3. Inconsistency test.**

**Assessment of publication bias and small-sample effect detection.** Fig 6 shows the comparison-correction funnel plots of the included comparison. The funnel diagram is basically

**Table 3. Node analysis.**

| Side | Direct | | Indirect | | Difference | | P>z |
|---|---|---|---|---|---|---|---|
| | Coef. | Std. Err. | Coef. | Std. Err. | Coef. | Std. Err. | |
| A H | 0.863 | 0.429 | -1.012 | 42.161 | 1.874 | 42.163 | 0.965 |
| B E | -2.123 | 2.001 | 5.147 | 3110.084 | -7.269 | 3110.085 | 0.998 |
| B H | 2.605 | 1.47 | 3.023 | 0.637 | -0.417 | 1.602 | 0.794 |
| B I | 1.852 | 0.218 | 1.435 | 1.587 | 0.417 | 1.602 | 0.794 |
| C F | 1.899 | 0.239 | 6.653 | 203.357 | -4.754 | 203.357 | 0.981 |
| D G | 1.441 | 0.314 | 1.876 | 0.905 | -0.435 | 0.958 | 0.65 |
| D I | 0.989 | 0.297 | 0.555 | 0.911 | 0.435 | 0.958 | 0.65 |
| F G | -3.269 | 0.974 | -2.678 | 0.465 | -0.591 | 1.061 | 0.578 |
| F H | -1.791 | 0.631 | -3.259 | 0.998 | 1.468 | 1.181 | 0.214 |
| F I | -3.377 | 0.237 | -2.351 | 1.068 | -1.026 | 1.102 | 0.352 |
| G H | -0.773 | 1.229 | 1.039 | 0.724 | -1.812 | 1.426 | 0.204 |
| G I | -0.339 | 1.418 | -0.556 | 0.401 | 0.218 | 1.473 | 0.882 |

*Node analysis results*

**Table 4. Network meta-analysis result.**

| | | | | | | | | |
|---|---|---|---|---|---|---|---|---|
| postnatal 3/I | 3.04 (1.02,9.03) | 1.72 (0.81,3.66) | 27.71 (17.66,43.50) | 0.02 (0.00,0.98) | 0.39 (0.22,0.67) | 4.15 (2.17,7.94) | 0.16 (0.10,0.24) | 1.28 (0.32,5.08) |
| 0.33 * (0.11,0.98) | postnatal 2/H | 0.56 (0.17,1.92) | 9.11 (3.20,25.92) | 0.01 (0.00,0.37) | 0.13 (0.04,0.41) | 1.36 (0.43,4.29) | 0.05 (0.02,0.16) | 0.42 (0.18,0.98) |
| 0.58 (0.27,1.24) | 1.77 (0.52,6.02) | postnatal 1/G | 16.14 (7.00,37.22) | 0.01 (0.00,0.61) | 0.23 (0.13,0.40) | 2.42 (0.93,6.29) | 0.09 (0.04,0.22) | 0.75 (0.17,3.30) |
| 0.04 * (0.02,0.06) | 0.11* (0.04,0.31) | 0.06 * (0.03,0.14) | Blank/F | 0.00 (0.00,0.04) | 0.01 (0.01,0.03) | 0.15 (0.09,0.24) | 0.01 (0.00,0.01) | 0.05 (0.01,0.18) |
| 52.84 * (1.02,2730.18) | 160.72 * (2.70,9562.99) | 90.71 * (1.64,5031.73) | 1464.44 * (27.65,77566.59) | antenatal 5/E | 20.48 (0.38,1099.57) | 219.16 (4.03,11931.17) | 8.35 (0.17,421.95) | 67.86 (1.05,4398.56) |
| 2.58 * (1.48,4.48) | 7.85 * (2.44,25.26) | 4.43 * (2.48,7.92) | 71.49 * (35.95,142.18) | 0.05 (0.00,2.62) | antenatal 4/D | 10.70 (4.66,24.57) | 0.41 (0.20,0.82) | 3.31 (0.78,13.98) |
| 0.24 * (0.13,0.46) | 0.73 (0.23,2.31) | 0.41 (0.16,1.08) | 6.68 * (4.19,10.67) | 0.00 * (0.00,0.25) | 0.09 * (0.04,0.21) | antenatal 3/C | 0.04 (0.02,0.08) | 0.31 (0.07,1.28) |
| 6.33 * (4.15,9.65) | 19.24 * (6.12,60.49) | 10.86 * (4.58,25.73) | 175.29 * (94.99,323.46) | 0.12 (0.00,6.05) | 2.45 * (1.22,4.91) | 26.23 * (12.14,56.70) | antenatal 2/B | 8.12 (1.96,33.64) |
| 0.78 (0.20,3.08) | 2.37* (1.02,5.49) | 1.34 (0.30,5.89) | 21.58 * (5.64,82.53) | 0.01 * (0.00,0.96) | 0.30 (0.07,1.27) | 3.23 (0.78,13.37) | 0.12 (0.03,0.51) | antenatal 1/A |

* indicates a significant difference in the data (*P<0.05*).

symmetric, and the regression line is less tilted; therefore, this study may have a small sample effect and slight publication bias.

## Discussion

In 2012, the National Institute of Health and Clinical Optimization (NICE) proposed that preventing maternal antibody sensitization via routine prenatal anti-D prophylaxis (RAADP) is a cost-effective method (http://www.nice.org.uk/). The British Committee for Standards in Haematology (BCSH) published the latest guidelines in 2014, recommending the use of anti-D immunoglobulin to prevent haemolytic disease of the fetus and newborn [32]. These guidelines recommend that RAADP be performed in RhD-negative pregnant women in their third trimester of pregnancy. RAADP includes the following regimens: a single dose of 300 μg (1500 IU) anti-D immunoglobulin between 28 and 30 gestational weeks or a two-dose regimen of 100 μg (500 IU) anti-D immunoglobulin at 28 and 34 gestational weeks. A Kleihauer-Betke test should be performed after delivery to estimate whether fetomaternal hemorrhaging exceeded 4 ml, then another 100ug(500 IU) should be administered within 72 hours of delivery. In 2017, the American College of Obstetricians and Gynecologists (ACOG)[33] recommends prophylactic anti-D immune globulin to unsensitized Rh D-negative women at 28 weeks of gestation. After birth, if the baby is Rh D positive, these mothers should receive anti-D immune globulin within 72 hours of birth.

However, an observational study noted that compliance with the single injection regimen was better than that with the two-injection regimen[34]. A single injection may also reduce costs. No evidence exists to assess the efficacy of these therapeutic strategies[32].

Therefore, we conducted a network meta-analysis comparing multiple treatment measures. The results revealed that the antenatal 5/E, antenatal 2/B, antenatal 4/D, antenatal 1/A, postnatal 3/I, postnatal 2/H and antenatal 3/C regimens significantly reduced serum anti-D antibody positive rates compared with that of the blank/F regimen alone, indicating that anti-D immunoglobulin immunotherapy, whether administered before or after delivery, significantly

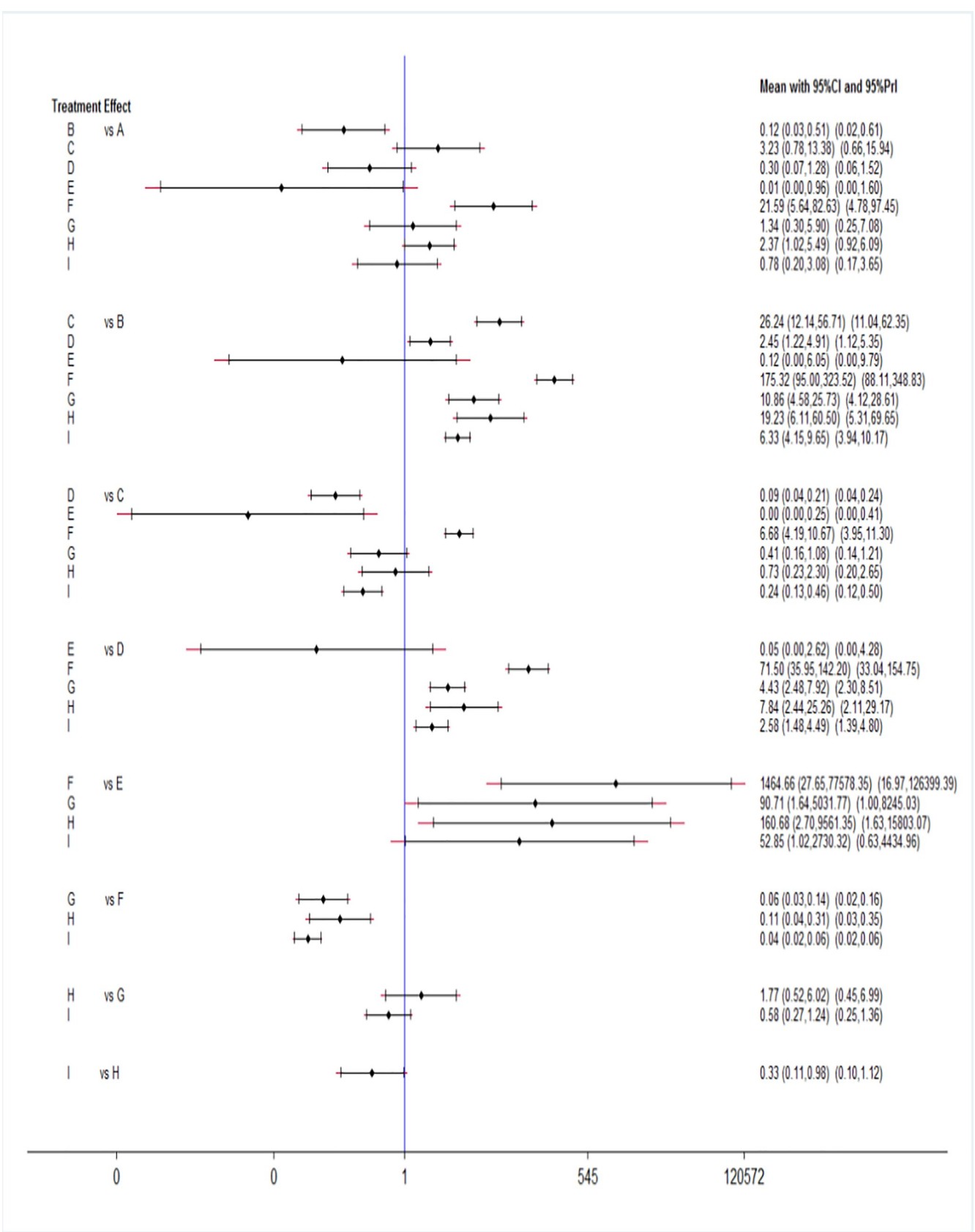

**Fig 4. Forest plot of the network meta-analysis.**

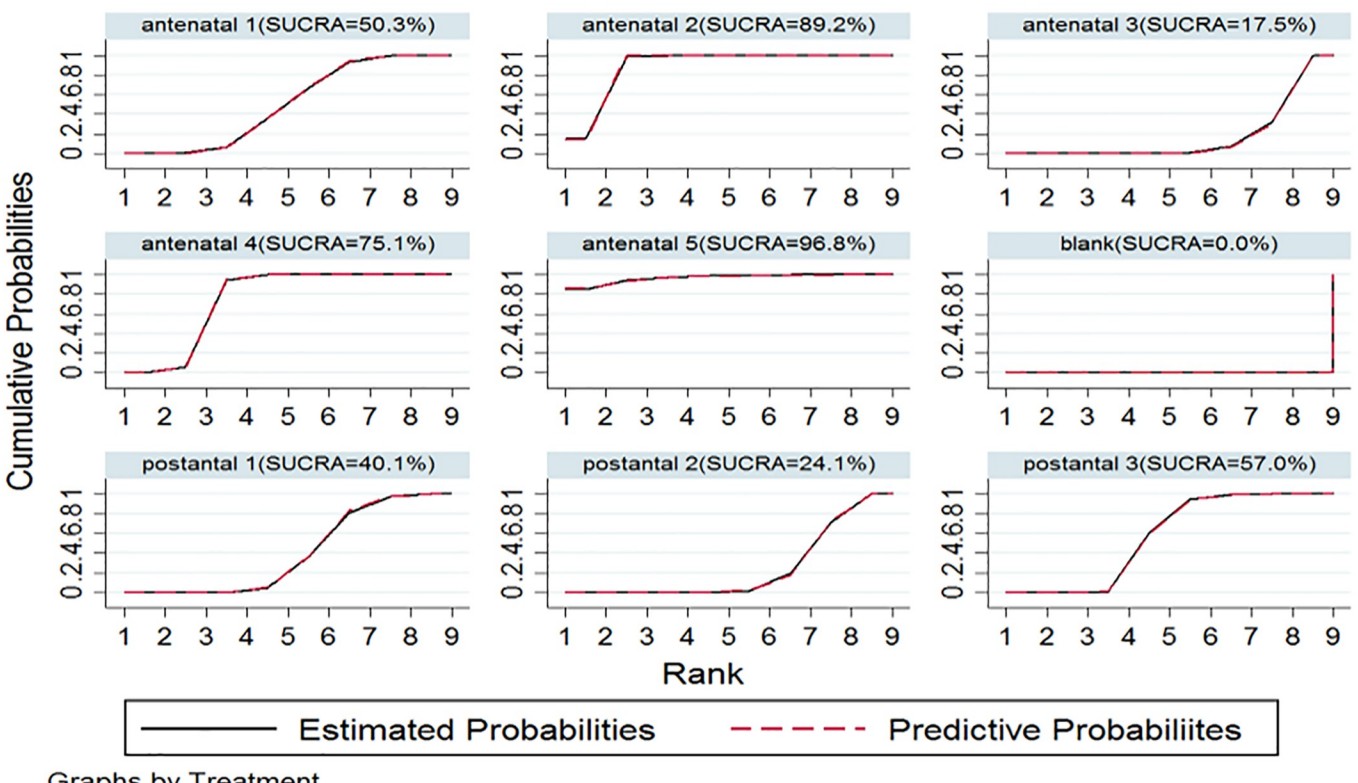

**Fig 5. SUCRA for preventing maternal antibody sensitization.**

reduced the incidence of maternal serum anti-D antibody positivity in Rh-negative mothers with Rh-positive fetuses. Moreover, antenatal 5, antenatal 2, antenatal 4, antenatal 1 and postnatal 3 were the most effective treatment measures for reducing the incidence of maternal anti-D antibody positivity. Therapeutic regimens antenatal 5 and antenatal 2 were likely the most effective regimens for preventing hemolytic diseases in fetuses and newborns.

The SUCRA for preventing maternal antibody sensitization indicated that the 300-μg anti-D immunoglobulin injection at 28 and 34 gestational weeks (antenatal 5/E) was likely the most effective regimen (SUCRA = 96.8%), and administering 300 μg within 28 gestational weeks (antenatal 2/B) was likely the second most effective (SUCRA = 89.2%). Administering no injection or a placebo was the least effective regimen (SUCRA = 0.0%). The anti-D immunoglobulin mechanism of action, which is closely related to the drug duration and dose, may explain these results. Anti-D immunoglobulin is extracted from the serum and used to prevent neonatal hemolysis.

RhD-positive red blood cells (containing the D antigen) from the fetus stimulate antibody production in RhD-negative mothers. During pregnancy and delivery of the first RhD-positive fetus to RhD-negative mothers, the red blood cells of the RhD-positive fetuses enter the RhD-negative mothers and stimulate the mothers to produce IgG anti-D antibodies. When an RhD-negative mother later carries an RhD-positive fetus, the antibodies in the maternal serum enter the fetal blood circulation via the placental barrier and can cause neonatal hemolysis.

However, during the pregnancy with the first RhD-positive fetus, or within 72 hours after delivery, RhD-negative mothers can be intramuscularly injected with 300 μg anti-D

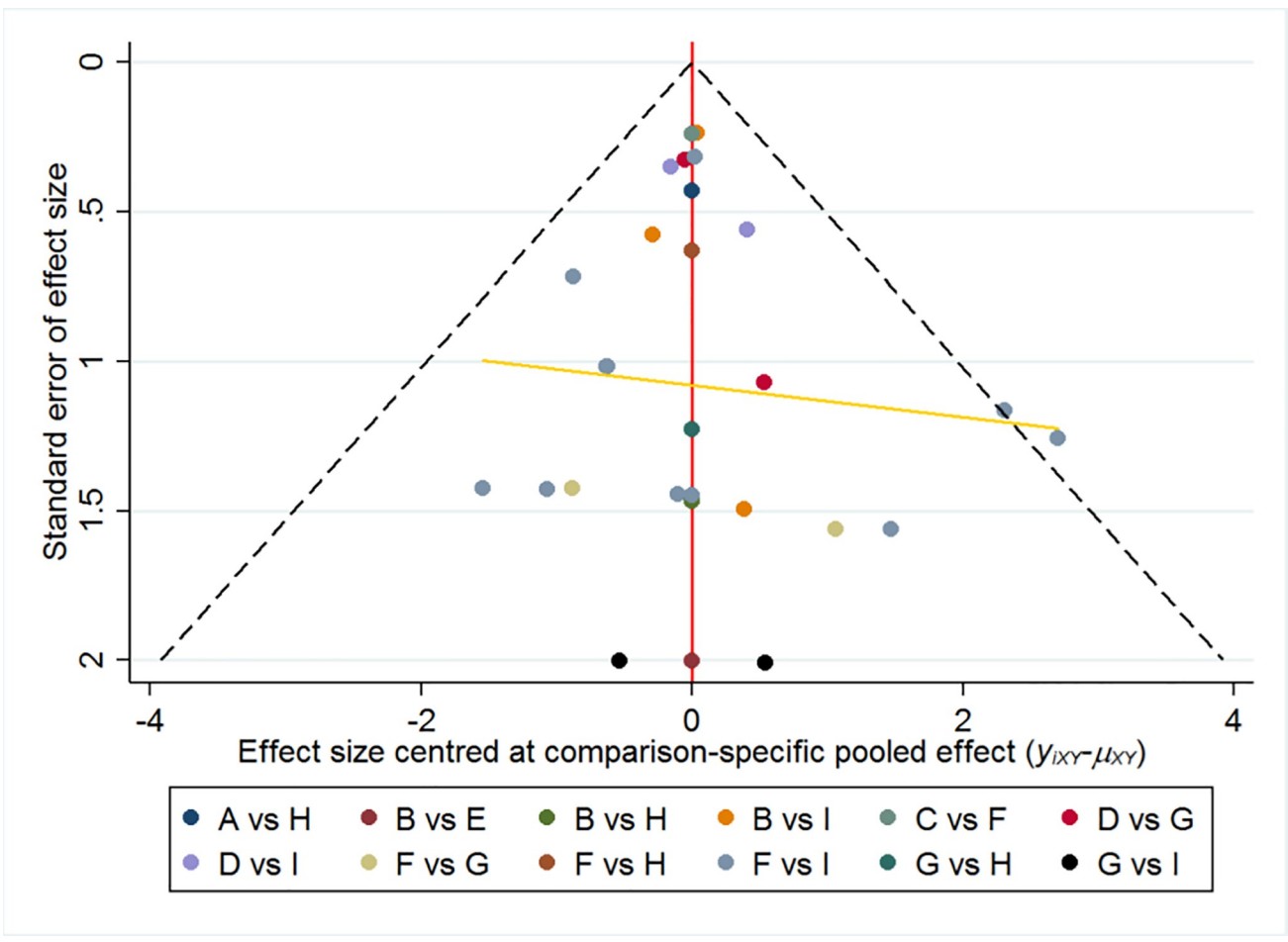

**Fig 6. Correction funnel plot.**

immunoglobulin, which can bind to the D antigen leaked into the mother's serum and desensitize it, thus blocking anti-D antibody production in the mother's serum. Anti-D immunoglobulin had no significant preventive effect on mothers who had already produced anti-D antibodies.

Twenty-five micrograms (125 IU) of anti-D immunoglobulin can typically protect against a fetal-maternal hemorrhage (FMH) of approximately 1–2 ml of blood. Therefore, 100 µg (500 IU) of anti-D antibody can prevent an FMH of approximately 8 ml, and 300 µg can prevent an FMH of approximately 30 ml. An FMH of greater than 30 ml is uncommon [35]. However, pharmacokinetic studies have shown that anti-RhD levels vary among patients, and some may have insufficient anti-RhD levels during childbirth [36]. A single injection of 300 µg anti-D immunoglobulin maintained a high immunopreventive effect for approximately 12 weeks. Bowman et al. [37]suggested that women who failed to give birth within 12 weeks after receiving the prenatal doses should receive a second dose of anti-D immunoglobulin to maintain the immunopreventive effect.

Routine prenatal prophylaxis with anti-D immunoglobulin is unlikely to benefit the current pregnancy or improve pregnancy outcomes, but it can reduce the anti-D antibody production during subsequent pregnancies. Chilcott et al.[38]noted that routine anti-D immunoglobulin injections should prevent future hemolytic diseases in infants. In many countries, including

the United Kingdom and Australia, the guidelines recommend routine universal prenatal anti-D immunoglobulin prevention (http://www.ranzcog.edu.au/ and http://www.rcog.org.uk/). The incidence of D-negative individuals varies among ethnic groups, with the highest being in Basques (30% -35%), followed by North American and European Caucasians (15%) [38]. In China, RhD-negative individuals constitute approximately 0.3% of the population [39]. Routine use of anti-D immunoglobulin is the main method of decreasing the erythrocyte alloimmunity ratio.

## Conclusions

Several limitations must be considered when interpreting the results of this meta-analysis. First, the literature included in this study spanned a long time period, and the titer or quality of anti-D immunoglobulin varies over time, which may affect the outcome. Second, the recruited participants were all from western countries, and no studies could be found regarding Asians and anti-D immunoglobulin. This might limit the application of our conclusions, and research on other races should be conducted.

In conclusion, this study showed that the current first-line recommendation is two 300-μg prenatal immunizations at 28 and 34 gestational weeks. If the anti-D immunoglobulin supply is inadequate, the second alternative should be a single 300-μg prenatal immunization at 28 gestational weeks.

## Supporting information

**S1 Checklist. PRISMA NMA checklist of Items to include when reporting a systematic review involving a network meta-analysis.**
(DOCX)

**S1 Table. The raw data of all included studies.**
(XLSX)

## Author Contributions

**Data curation:** Xiaohui Xie, Qiurong Fu.

**Formal analysis:** Xiaohui Xie, Yi Zhang, Dan Zhou.

**Investigation:** Yi Zhang.

**Resources:** Xiaohui Xie, Qiurong Fu, Ziwei Bao, Dan Zhou.

**Software:** Xiaohui Xie, Yi Zhang, Dan Zhou.

**Writing – original draft:** Xiaohui Xie, Yi Zhang, Dan Zhou.

**Writing – review & editing:** Xiaohui Xie, Ziwei Bao, Yi Zhang, Dan Zhou.

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
