## [Decision Letter · Decision Letter 0]

28 Jan 2020

PONE-D-19-24055

Clinical value of different anti-D immunoglobulin strategies for preventing Rh hemolytic disease of the fetus and newborn: A network meta-analysis

PLOS ONE

Dear Mr Zhang,

Thank you for submitting your manuscript to PLOS ONE. After careful consideration, we feel that it has merit but does not fully meet PLOS ONE’s publication criteria as it currently stands. There were several major comments and concerns raised during the review of your manuscript. Therefore, we invite you to submit a revised version of the manuscript that addresses ALL of the points raised during the review process.

We would appreciate receiving your revised manuscript by Mar 12 2020 11:59PM. To enhance the reproducibility of your results, we recommend that if applicable you deposit your laboratory protocols in protocols.io, where a protocol can be assigned its own identifier (DOI) such that it can be cited independently in the future. For instructions see: http://journals.plos.org/plosone/s/submission-guidelines#loc-laboratory-protocols

We look forward to receiving your revised manuscript.

Kind regards,

Frank T. Spradley

Academic Editor

PLOS ONE

2. Please upload a new copy of Figure 2 as the detail is not clear. Please follow the link for more information: http://blogs.PLOS.org/everyone/2011/05/10/how-to-check-your-manuscript-image-quality-in-editorial-manager/

3. Please include your tables as part of your main manuscript and remove the individual files. Please note that supplementary tables (should remain/ be uploaded) as separate "supporting information" files

4. . We note you have included two different tables in your manuscript labelled as Table 3; 'Table 3 Treatment abbreviations' and 'Table 3 Network meta-analysis result'. Please review these and ensure that you refer to all tables in the text of your manuscript so that they can be separately identified; if accepted, production will need this reference to link the reader to each Table.

Reviewers' comments:

Reviewer's Responses to Questions

**Comments to the Author**

1. Is the manuscript technically sound, and do the data support the conclusions?

Reviewer #1: Yes

2. Has the statistical analysis been performed appropriately and rigorously? 

Reviewer #1: Yes

3. Have the authors made all data underlying the findings in their manuscript fully available?

Reviewer #1: Yes

4. Is the manuscript presented in an intelligible fashion and written in standard English?

Reviewer #1: Yes

5. Review Comments to the Author

Reviewer #1: The authors investigated the various anti-d immunoglobulin strategies used for prevention of Rh hemolytic disease of the fetus and newborn using a network meta-analysis. Of the nine therapeutic measures studied, they found that giving 300 µg anti-D immunoglobulin at 28 and 34 weeks of gestation was the most effective measure for preventing maternal antibody sensitization, followed by a single injection at 28 weeks of gestation. This is interesting work clearly written.

Reviewer’s comments;

1. Page 3 (Results, last sentence): Suggest adding - Table 2 has the treatment abbreviations.

2. Page 5 (Table 2): This table was incorrectly labeled as Table 3. Please correct.

3. Page 9 (Discussion, middle of first paragraph): Suggest adding 100 µg to the sentence that states that …then another 500 IU should be administered…

4. Page 9 (Discussion, first paragraph): ACOG is the American College of Obstetricians and Gynecologists. Please correct.

5. Page 9 (Discussion, first paragraph): The sentence …(ACOG) recommended a single injection of 300 µg anti-D immunoglobulin at 28 weeks of pregnancy with an RhD–positive baby, then another injection of 300 µg again after birth… should be corrected as it is confusing as written. ACOG recommends prophylactic anti-D immune globulin to unsensitized Rh D-negative women at 28 weeks of gestation. After birth, if the baby is Rh D positive, these mothers should receive anti-D immune globulin within 72 hours of birth.

6. Page 10 (Discussion, top paragraph, first line): The area in parenthesis after …anti-D antibody positive rates compared with that of the blank/F regimen alone… can be deleted as it already appears in the Results section.

7. Page 10 (Discussion, top paragraph, line 10): Similarly as in #6 above, the area in parenthesis after …incidence of maternal anti-D positivity…can be deleted.

8. Page 11 (Discussion, next to last paragraph, last four lines): Please clarify what you mean by …but few studies have focused on the first fetus born to RhD-negative mothers. Supply references of the few studies. Also, please reference the last sentence …Some studies have also found that using anti-D immunoglobulin may cause hemolysis in fetuses during pregnancy.

6. PLOS authors have the option to publish the peer review history of their article (what does this mean?). If published, this will include your full peer review and any attached files.

Reviewer #1: No

---

## [Author Response · Author response to Decision Letter 0]

1 Feb 2020

Dear reviewers:

I am very grateful to your comments for the manuscript. According with your advice, we amended the relevant part in manuscript. Some of your questions were answered below.

 The authors’ answer: the manuscript has been revised as PLOS ONE's style requirements

2. Please upload a new copy of Figure 2 as the detail is not clear. Please follow the link for more information: http://blogs.PLOS.org/everyone/2011/05/10/how-to-check-your-manuscript-image-quality-in-editorial-manager/

 The authors’ answer: The new Figure 2 has been repalced. 

3. Please include your tables as part of your main manuscript and remove the individual files. Please note that supplementary tables (should remain/ be uploaded) as separate "supporting information" files

 The authors’ answer: The tables has been inclued in the manuscript.There is no supplementary tables.

4. We note you have included two different tables in your manuscript labelled as Table 3; 'Table 3 Treatment abbreviations' and 'Table 3 Network meta-analysis result'. Please review these and ensure that you refer to all tables in the text of your manuscript so that they can be separately identified; if accepted, production will need this reference to link the reader to each Table.

 The authors’ answer: Table label has been modified .Table 3 Treatment abbreviations has been replaced by Table 2 Treatment abbreviations. 

Review Comments to the Author

Reviewer #1: The authors investigated the various anti-d immunoglobulin strategies used for prevention of Rh hemolytic disease of the fetus and newborn using a network meta-analysis. Of the nine therapeutic measures studied, they found that giving 300 µg anti-D immunoglobulin at 28 and 34 weeks of gestation was the most effective measure for preventing maternal antibody sensitization, followed by a single injection at 28 weeks of gestation. This is interesting work clearly written.

Reviewer’s comments;

1. Page 3 (Results, last sentence): Suggest adding - Table 2 has the treatment abbreviations.

The authors’ answer: The sentence has been added.

2. Page 5 (Table 2): This table was incorrectly labeled as Table 3. Please correct.

The authors’ answer: The Table label has been corrected.

3. Page 9 (Discussion, middle of first paragraph): Suggest adding 100 µg to the sentence that states that …then another 500 IU should be administered…

The authors’ answer: The “100 µg” has been added in the sentence.

4. Page 9 (Discussion, first paragraph): ACOG is the American College of Obstetricians and Gynecologists. Please correct.

The authors’ answer: “American Association of Obstetricians and Gynecologists” has been replaced by “American College of Obstetricians and Gynecologists”.

5. Page 9 (Discussion, first paragraph): The sentence …(ACOG) recommended a single injection of 300 µg anti-D immunoglobulin at 28 weeks of pregnancy with an RhD–positive baby, then another injection of 300 µg again after birth… should be corrected as it is confusing as written. ACOG recommends prophylactic anti-D immune globulin to unsensitized Rh D-negative women at 28 weeks of gestation. After birth, if the baby is Rh D positive, these mothers should receive anti-D immune globulin within 72 hours of birth.

The authors’ answer: The confusing sentence has been revised.

6. Page 10 (Discussion, top paragraph, first line): The area in parenthesis after …anti-D antibody positive rates compared with that of the blank/F regimen alone… can be deleted as it already appears in the Results section.

The authors’ answer: The area in parenthesis has been deleted.

7. Page 10 (Discussion, top paragraph, line 10): Similarly as in #6 above, the area in parenthesis after …incidence of maternal anti-D positivity…can be deleted.

The authors’ answer: The area in parenthesis has been deleted.

8. Page 11 (Discussion, next to last paragraph, last four lines): Please clarify what you mean by …but few studies have focused on the first fetus born to RhD-negative mothers. Supply references of the few studies. Also, please reference the last sentence …Some studies have also found that using anti-D immunoglobulin may cause hemolysis in fetuses during pregnancy.

The authors’ answer: Confused sentences have been deleted.

We acknowledge the reviewer’s comments and suggestions very much, which are valuable in improving the quality of our manuscript.

Sincerely yours,

Xiaohui Xie1;Yi Zhang2 

1.Department of Obstetrics and Gynecology，the First People's Hospital of Neijiang,Neijiang 641000,Sichuan Province, P. R. China2．Department of General Surgery，the First People's Hospital of Neijiang,Neijiang 641000,Sichuan Province, P. R. China

January 30, 2020

---

## [Decision Letter · Decision Letter 1]

13 Feb 2020

PONE-D-19-24055R1

Clinical value of different anti-D immunoglobulin strategies for preventing Rh hemolytic disease of the fetus and newborn: A network meta-analysis

PLOS ONE

Dear Mr Zhang,

Thank you for submitting your manuscript to PLOS ONE. After careful consideration, we feel that it has merit but does not fully meet PLOS ONE’s publication criteria as it currently stands. There are two additional comments from the reviewer that must be addressed. Therefore, we invite you to submit a revised version of the manuscript that addresses the points raised during the review process.

We would appreciate receiving your revised manuscript by Mar 29 2020 11:59PM. To enhance the reproducibility of your results, we recommend that if applicable you deposit your laboratory protocols in protocols.io, where a protocol can be assigned its own identifier (DOI) such that it can be cited independently in the future. For instructions see: http://journals.plos.org/plosone/s/submission-guidelines#loc-laboratory-protocols

We look forward to receiving your revised manuscript.

Kind regards,

Frank T. Spradley

Academic Editor

PLOS ONE

Reviewers' comments:

Reviewer's Responses to Questions

**Comments to the Author**

1. If the authors have adequately addressed your comments raised in a previous round of review and you feel that this manuscript is now acceptable for publication, you may indicate that here to bypass the “Comments to the Author” section, enter your conflict of interest statement in the “Confidential to Editor” section, and submit your "Accept" recommendation.

Reviewer #1: (No Response)

2. Is the manuscript technically sound, and do the data support the conclusions?

Reviewer #1: Yes

3. Has the statistical analysis been performed appropriately and rigorously? 

Reviewer #1: Yes

4. Have the authors made all data underlying the findings in their manuscript fully available?

Reviewer #1: Yes

5. Is the manuscript presented in an intelligible fashion and written in standard English?

Reviewer #1: Yes

6. Review Comments to the Author

Reviewer #1: The investigators have addressed previous comments and have made changes to the manuscript. However, I have two additional comments:

1. Results of the Bayesian network meta-analysis (lines 6-7): Based on Table 4, the results for postnatal 3/I vs blank/F should be OR=0.04, 95% CI=0.02-0.06. Please clarify.

2. Figure 5: The labeling of the various curves should be reevaluated. For example, antenatal 5/E had a SUCRA of 96.8%, however the Figure has antenatal 1 with that SUCRA. Antenatal 2/B had a SUCRA of 89.2%, however the Figure has antenatal 3 with that SUCRA.

7. PLOS authors have the option to publish the peer review history of their article (what does this mean?). If published, this will include your full peer review and any attached files.

Reviewer #1: No

---

## [Author Response · Author response to Decision Letter 1]

18 Feb 2020

Reviewer #1: The investigators have addressed previous comments and have made changes to the manuscript. However, I have two additional comments:

1. Results of the Bayesian network meta-analysis (lines 6-7): Based on Table 4, the results for postnatal 3/I vs blank/F should be OR=0.04, 95% CI=0.02-0.06. Please clarify.

Answer: the results for postnatal 3/I vs blank/F has been revised.

2. Figure 5: The labeling of the various curves should be reevaluated. For example, antenatal 5/E had a SUCRA of 96.8%, however the Figure has antenatal 1 with that SUCRA. Antenatal 2/B had a SUCRA of 89.2%, however the Figure has antenatal 3 with that SUCRA.

 Answer:The labeling of the various curves has been reevaluated.The Figure 5 has been replaced.

Supporting Information files and PRISMA checklist are uploaded.please check

---

## [Decision Letter · Decision Letter 2]

21 Feb 2020

Clinical value of different anti-D immunoglobulin strategies for preventing Rh hemolytic disease of the fetus and newborn: A network meta-analysis

PONE-D-19-24055R2

Dear Dr. Zhang,

We are pleased to inform you that your manuscript has been judged scientifically suitable for publication and will be formally accepted for publication once it complies with all outstanding technical requirements.

With kind regards,

Frank T. Spradley

Academic Editor

PLOS ONE

Reviewers' comments:

Reviewer's Responses to Questions

**Comments to the Author**

1. If the authors have adequately addressed your comments raised in a previous round of review and you feel that this manuscript is now acceptable for publication, you may indicate that here to bypass the “Comments to the Author” section, enter your conflict of interest statement in the “Confidential to Editor” section, and submit your "Accept" recommendation.

Reviewer #1: All comments have been addressed

2. Is the manuscript technically sound, and do the data support the conclusions?

Reviewer #1: Yes

3. Has the statistical analysis been performed appropriately and rigorously? 

Reviewer #1: Yes

4. Have the authors made all data underlying the findings in their manuscript fully available?

Reviewer #1: Yes

5. Is the manuscript presented in an intelligible fashion and written in standard English?

Reviewer #1: Yes

6. Review Comments to the Author

Reviewer #1: (No Response)

7. PLOS authors have the option to publish the peer review history of their article (what does this mean?). If published, this will include your full peer review and any attached files.

Reviewer #1: No

---

## [Editor Report · Acceptance letter]

26 Feb 2020

PONE-D-19-24055R2 

Clinical value of different anti-D immunoglobulin strategies for preventing Rh hemolytic disease of the fetus and newborn: A network meta-analysis 

Dear Dr. Zhang:

I am pleased to inform you that your manuscript has been deemed suitable for publication in PLOS ONE. Congratulations! Your manuscript is now with our production department. 

With kind regards,

on behalf of

Dr. Frank T. Spradley 

Academic Editor

PLOS ONE